

# A new software toolkit for optical apportionment of carbonaceous aerosol

Tommaso Isolabella[1,2], Vera Bernardoni[3], Alessandro Bigi[4], Marco Brunoldi[1,2], Federico Mazzei[1,2], Franco Parodi[2], Paolo Prati[1,2], Virginia Vernocchi[2], Dario Massabò[1,2]

[1] Physics Department, University of Genoa, 16146 Genoa, Italy
[2] I.N.F.N., Division of Genoa, 16146 Genoa, Italy
[3] Dipartimento di Fisica, Università degli Studi di Milano, and I.N.F.N. Division of Milan, 20133 Milan, Italy
[4] Department of Engineering 'Enzo Ferrari', University of Modena and Reggio Emilia, 41125 Modena, Italy

*Correspondence to*: Federico Mazzei (federico.mazzei@ge.infn.it) - Dario Massabò (dario.massabo@ge.infn.it)

**Abstract.** Instruments measuring aerosol light absorption, such as the Aethalometer and the Multi-Wavelength Absorbance Analyzer (MWAA), have been extensively used to characterize optical absorption of atmospheric particulate matter. Data retrieved with such instruments can be analysed with mathematical models to apportion different aerosol sources (Aethalometer model) and components (MWAA model). In this work we present an upgrade to the MWAA optical apportionment model. In addition to the apportionment of the absorption coefficient $b_{abs}$ in its components (Black Carbon and Brown Carbon) and sources (Fossil Fuel and Wood Burning), the extended model allows the retrieval of the Absorption Ångström Exponent of each component and source, thereby avoiding initial assumptions regarding these parameters. We also present a new open-source software toolkit, the MWAA Model Toolkit, written in both Python and R, that performs the entire apportionment procedure.

## 1 Introduction

Atmospheric Particulate Matter (PM) plays an important role in environmental issues such as human health, air quality and climate change (Seinfeld and Pandis, 2016). Several chemical species and aggregates, present in the atmosphere, affect the energy balance of the Earth system by absorbing and scattering solar radiation (Laj et al., 2020). A variety of sources contribute to the emission of light absorbing or scattering PM: their identification and quantification are necessary to mitigate the harmful effects of PM, especially in the climate change issue.

Between other constituents, Black Carbon (BC) and Brown Carbon (BrC) are the most light-absorbing components of PM (Bond et al., 2013). BC consists of fractal-like chains of submicron particles, and it is formed by incomplete combustion processes. Due to the wavelength independence of the imaginary part of its refractive index, it is a strong light absorber across the entire visible range. BrC represents a more elusive class of organic carbonaceous compounds whose defining characteristic is to absorb radiation more efficiently at shorter visible bands than at longer wavelengths, where its absorption is considered negligible (Poeschl, 2003; Andreae and Gelencser, 2006). The composition of BrC is still poorly understood, due to its chemical complexity and spatiotemporal variability; it consists of a number of molecular weight compounds, generally prone to oxidation and chemically unstable (Forrister et al., 2015). BrC is emitted directly through combustion of biomass but can also be formed as a product of secondary processes in the atmosphere (Liu et al., 2015, Tang et al., 2016).

Other aerosol compounds exhibit strong, albeit more selective, light-interaction properties. One such example is mineral dust, which is the most widespread aerosol type in terms of total mass, with a consequent important impact on the Earth's



energy balance due to its light absorption properties (Alfaro et al., 2009, Caponi et al., 2015 Schepanski, 2018, Di Biagio et al, 2019). However, in this work, we restrict our attention to carbonaceous aerosol and its sources.

In general, the spectral dependence of light absorption by small particles can be parameterized with a power-law function. In particular, the aerosol absorption coefficient $b_{abs}$ can be written as a function of the wavelength of the incoming radiation as $b_{abs}(\lambda) = c\lambda^{-\alpha}$, where $c$ is a proportionality factor and $\alpha$, the Absorption Ångström Exponent (AAE), defines

the spectral dependence of the absorption. Different aerosol types correspond to different values of $\alpha$, which has been shown to depend on particle size, morphology, chemical composition, and mixing/ageing state (Moosmüller et al 2011; Utry et al 2014). For BC in its ideal form (spherical particles with no wavelength-dependence of the imaginary part of the refractive index), the literature is consistent in indicating the $\alpha$ value of 1, both for real-world (Bond and Bergstrom, 2006) and produced in controlled conditions (Vernocchi et al., 2022) samples, whereas much more variation is encountered in

the value of $\alpha$ for BrC, with reported values ranging up to 9.5 (Hoffer et al., 2006; Harrison et al., 2013; Lack and Langridge; 2013). This is likely due to the broader range of chemical composition and effects of ageing. Intermediate values of $\alpha$ are observed for aerosols containing both BC and BrC (Massabò et al., 2019). This significant difference in the wavelength-dependent behaviour of light-absorbing components can be used as an efficient tool for the source and component apportionment of light-absorbing aerosol.

Source apportionment models exploiting the power-law behaviour of $b_{abs}$ have been successfully applied to multi-$\lambda$ measurements of absorption. The Aethalometer model (Sandradewi et al., 2008), allows the apportionment of the absorption coefficient to two different sources, namely Fossil Fuel (FF) and Wood Burning (WB), exploiting the different $\alpha$ that characterizes the aerosol produced by the two sources. The MWAA (Multi-Wavelength Absorbance Analyzer) model extends the Aethalometer model by explicitly including the apportionment of optical absorption due to BC and

BrC, resulting in an algorithm that allows the differentiation of both aerosol sources and components, based on at least $5 - \lambda$ absorption measurements (Massabò et al., 2015; Bernardoni et al., 2017). Both the Aethalometer model and the MWAA model are effective in apportioning aerosol absorption, but they have a conceptual drawback: the values of some physical parameters must be fixed prior to the analysis in order to run the algorithm. These parameters are the $\alpha$ for FF and WB ($\alpha_{FF}$ and $\alpha_{WB}$ for the Aethalometer model, and, in addition, $\alpha_{BC}$ for the MWAA model). Since the $\alpha$ depends

on a variety of factors, as mentioned above, fixing these exponents for the analysis, according to the literature, can lead to errors, since the actual value of these exponents may be different for the specific aerosol analysed. The only way to avoid this problem is to retrieve these crucial parameters by using information obtained by independent techniques/methods (e.g., Levoglucosan, [14]C, receptor models, others), as stated in several recent publications in the literature (Massabò et al., 2015; Martinsson et al., 2017; Titos et al., 2017, Helin et al., 2018; Ivančič et al., 2022).

In this work we propose an upgrade to the original MWAA model that directly allows source and component apportionments of absorption data without the need to set any parameters before running the model. This is achieved by performing the apportionment analysis along with a correlation study with other, independent measurements of aerosol properties, such as chemical speciation or elemental composition. The parameters are then automatically set by the algorithm, based on the values that give the best correlation with the independent measurements. We also present a new

software toolkit written in two of the most widely used scientific programming languages, Python and R, to perform this analysis automatically. Output of the presented toolkit (MWAA_MT) are the following quantities: $\alpha_{FF}$, $\alpha_{WB}$, $\alpha_{BC}$, $\alpha_{BrC}$, and the carbonaceous masses for fossil fuels and wood burning: EC_FF/OC_FF, and EC_WB/OC_WB, respectively. Finally, to demonstrate the capability of the upgraded model, we provide an example application to data published elsewhere (Bernardoni et al., 2017).



## 2 Model description

The MWAA model has been extensively described elsewhere (Massabò et al., 2015). Here we will only report the main points to establish the notation and describe the upgrades.

The measured aerosol absorption coefficient $b_{abs}$ at different wavelengths is decomposed in two different ways:

$$b_{abs}(\lambda) = b_{abs}^{BC}(\lambda) + b_{abs}^{BrC}(\lambda) = A\lambda^{-\alpha_{BC}} + B\lambda^{-\alpha_{BrC}} \qquad (1)$$

and

$$b_{abs}(\lambda) = b_{abs}^{FF}(\lambda) + b_{abs}^{WB}(\lambda) = A'\lambda^{-\alpha_{FF}} + B'\lambda^{-\alpha_{WB}} \qquad (2)$$

Equation (1) represents the decomposition of the measured aerosol absorption coefficient, at each wavelength, into its contributions due to carbonaceous components, Black Carbon (BC) and Brown Carbon (BrC). Both species are assumed to absorb radiation according to a negative power law $b_{abs}(\lambda) \propto \lambda^{-\alpha}$, with a different absorption exponent for BC ($\alpha_{BC}$) and BrC ($\alpha_{BrC}$). In addition, the model assumes that BrC is only produced by wood burning.

Equation (2) has the same structure as the Aethalometer apportionment model (Sandradewi 2008), whereby the absorption coefficient is decomposed into contributions from different sources, namely Fossil Fuel (FF) and Wood Burning (WB). As in Eq. (1), these terms are also assumed to contribute to the total optical absorption following negative power law whose exponents are different for FF ($\alpha_{FF}$) and WB ($\alpha_{WB}$).

The parameters $A, B, A'$ and $B'$ are scaling factors proportional to the Mass Absorption Cross-section (MAC) of each component. In the original MWAA model, all but one of the exponents ($\alpha_{BrC}$) are fixed to appropriate values according to the literature (Sandradewi et al., 2008; Favez et al., 2010; Herich et al., 2011: Harrison et al., 2013; Massabò et al., 2015; Zotter et al., 2017; Forello et al., 2019), most commonly $\alpha_{BC} = \alpha_{FF} = 1$ and $\alpha_{WB} = 1.8$ or 2. Then, the multi-$\lambda$ measurements of $b_{abs}$ are fitted using Eq. (1) and (2), obtaining $A, B, A', B'$ and $\alpha_{BrC}$. The contribution of the different sources and species to the optical absorption is obtained as follows:

$$b_{abs}^{BC,WB}(\lambda) = (A - A')\lambda^{-\alpha_{BC}}$$
$$b_{abs}^{BC,FF}(\lambda) = A'\lambda^{-\alpha_{BC}} \qquad (3)$$
$$b_{abs}^{BrC}(\lambda) = B\lambda^{-\alpha_{BrC}}$$

The upgraded model we present eliminates the need to arbitrarily specify $\alpha_{BC}, \alpha_{FF}$ and $\alpha_{WB}$ by instead adjusting their values so that the apportioned contributions found in (3) have the best correlation with independent measurements. Figure 1 shows a streamlined version of the upgraded MWAA model, in which the independent measurement for the adjustment of the exponents is the levoglucosan content in the sample, as determined by chromatography. Levoglucosan is a strong tracer for biomass burning (Simoneit et al., 1999) and should therefore correlate well with $b_{abs}^{BrC}$ and $b_{abs}^{BC,WB}$.

The three parameters $\alpha_{BC}, \alpha_{FF}$ and $\alpha_{WB}$ are each varied within their range, while the others are held constant. In the first step, $\alpha_{BC}$ is varied in the set $\{0.8, 0.9, 1.0, 1.1, 1.2\}$. For each $\alpha_{BC}$ value, Eq. (1) is used to fit the data and $b_{abs}^{BrC}$ is calculated for the shortest available wavelength, using Eq. (3). The coefficient of determination $R^2$ for a linear regression between $b_{abs}^{BrC}$ and the levoglucosan concentration for all samples is calculated, and the $\alpha_{BC}$ value that maximizes $R^2$ is selected. In the second step, $\alpha_{WB}$ is set to 2 and a similar procedure as described for the first step is performed for on $\alpha_{FF}$. Finally, in the third step, the procedure is repeated to find a best value for $\alpha_{WB}$. It is worth noting that the permutation of the steps to minimize $\alpha_{FF}$ and $\alpha_{WB}$ leads to the same preprocessing results. These three steps are repeated $N = 3$ times, restricting



the range of variability for the parameters in each iteration to increase the accuracy of the search for the best values. To avoid statistically insignificant results, and to obtain a more robust result, a tolerance parameter $\Delta$ is introduced. If the increase of $R^2$ in each minimization routine is less than the tolerance, the previous value of the relevant $\alpha$ is retained.

In addition to the component and source apportionment of the optical absorption coefficient, the MWAA model provides a method to perform the apportionment of EC and OC masses to the fossil fuel and wood burning contributions:


$$EC_{FF} = \frac{b_{abs}^{BC,FF}(\lambda_l)}{b_{abs}(\lambda_l) - b_{abs}^{BrC}(\lambda_l)} \, EC$$

$$EC_{WB} = \frac{b_{abs}^{BC,WB}(\lambda_l)}{b_{abs}(\lambda_l) - b_{abs}^{BrC}(\lambda_l)} \, EC$$

$$EC = EC_{FF} + EC_{WB} \quad\quad (4)$$

$$OC_{FF} = k_1 b_{abs}^{BC,FF}(\lambda_l) + OC_{NC}$$


$$OC_{WB} = k_2 b_{abs}^{BrC}(\lambda_s) + OC_{NC}$$

$$OC = OC_{FF} + OC_{WB} + OC_{NC}$$

In the above equations, $\lambda_l$ and $\lambda_s$ represent the longest and shortest wavelengths for which a measurement is available, respectively; $OC_{NC}$ is the organic carbon produced by biogenic sources which is considered optically inactive; $k_1$ and $k_2$

are coefficients, in $g \, m^{-2}$, obtained by a linear regression of Eq. (4.3) and (4.4), in subsets of samples in which the OC concentration is low (for $k_1$) and high (for $k_2$). The coefficients $k_1$ and $k_2$ are related to the Mass Absorption Cross-sections (MACs) of BC and BrC, respectively, and to the ratios $OC_{FF}/BC_{FF}$ and $OC_{WB}/BrC$. For more details on the mass apportionment procedure, see Massabò et al., 2015.

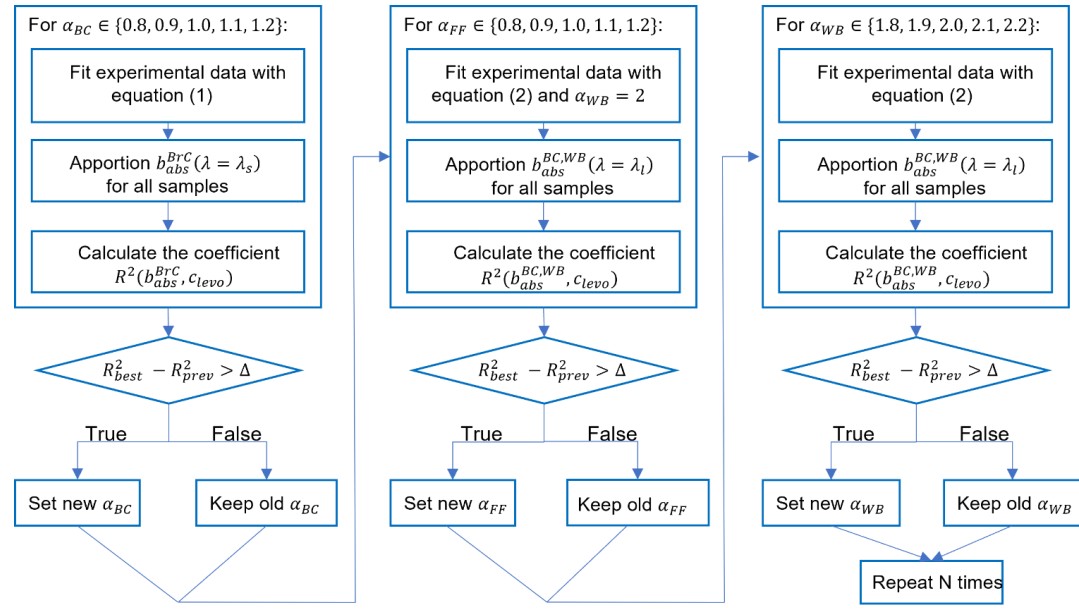


*Figure 1: Flowchart illustrating the pre-processing step in the improved MWAA model. $c_{levo}$ is the levoglucosan concentration in the samples, $R^2$ is the coefficient of determination in the linear regression, and the subscripts best and prev refer to, respectively,*



*the best correlation in the present iteration, and the best correlation in the previous iteration relative to the same α; Δ is a tuneable tolerance that prevents statistically insignificant fluctuations to assume a physical meaning.*

## 3 Software features

The software toolkit that performs the above-mentioned analysis has been released in the public domain. It can be installed via a standalone executable file (**download link**) or directly from source (**github link**). Currently, it can only be run in a Linux distribution (for example Ubuntu or Linux Mint).

MWAA_MT (the MWAA Model Toolkit) can perform optical and mass apportionment of data obtained with instruments measuring light absorption at least at five wavelengths, such as a multi $-\lambda$ Aethalometer.

The toolkit works in four separate steps:

Step I    It retrieves the best values for the three parameters $\alpha_{BC}, \alpha_{FF}$ and $\alpha_{WB}$ following the method detailed in the Sect. 2.

Step II   The values of $A, B, A', B'$ and $\alpha_{BrC}$ are obtained from fitting Eq. (1) and (2) with the remaining exponents fixed to the best values found in the previous step.

Step III  Equations (3) are employed to apportion the absorption coefficient at every wavelength.

Step IV   Following Eq. (4) the mass apportionment is performed for each sample.

The first step of the analysis is the most innovative aspect of the tool we introduce here, since the three values of
$\alpha_{BC}, \alpha_{FF}$ and $\alpha_{WB}$ are directly retrieved by the toolkit itself. However, if data from at least one independent analysis (e.g., Levoglucosan, [14]C, PMF, etc.) are not available, the user can still set these three parameters manually. In addition, the mass apportionment step is optional depending on the availability of carbonaceous mass measurements: if EC and OC are not available for the samples, the user can decide to skip the fourth step.

Similarly, the user can set many of the analysis hyperparameters. To perform the first step of the analysis, the current
version of the toolkit allows comparison of the optical apportionment with levoglucosan measurements, as described above. Future versions of MWAA_MT will likely allow the user to choose between different types of preprocessing analysis, considering different types of data such as [14]C measurements or source apportionment results obtained with independent techniques such as Positive Matrix Factorization or Multilinear Engine ME-2 receptor models.

Any number of samples can be provided for an analysis run; for the main optical apportionment procedure, steps II and
III, each sample is processed independently, whereas for the steps I and IV the entire dataset is considered for the regression analyses. Therefore, extra care must be taken to avoid entering obvious outliers as input to the software, and the analysis may need to be run twice, with the first run serving to weed out potential outliers and adjust the range of parameter variation.

## 4 Example of application: black and brown carbon optical apportionment of MWAA data

As an example of the application of MWAA_MT, we examine the results of the apportionment of two datasets previously published (Bernardoni et al., 2017). The first dataset is from a sampling campaign conducted in fall/winter 2014 in Propata, a rural site in northern Italy, while the second dataset is from a campaign conducted in winter 2016 at an urban background site in Milan, one of the largest cities in Italy. In these campaigns, PM10 samples were collected on quartz fibre filters, with each filter sampled (for 48h in Propata, for 12h in Milan) and then analysed with the MWAA instrument



to obtain the wavelength resolved absorption coefficients of the aerosol. All samples were analysed by liquid chromatography (HPLC-PAD) to determine the Levoglucosan concentration (Piazzalunga et al., 2010). Further details on the measurements can be found in Bernardoni et al. (2017). In the current study, we apply the updated MWAA model (MWAA_MT) to 28 samples from the Propata campaign (hereinafter referred to as "P" samples) and 20 samples from the Milan campaign ("AIN" samples). The aim of the comparison is to verify whether the particulate sampled in a rural

area has a different optical behaviour than the aerosol sampled in an urban area. The following steps were performed identically for both datasets.

### 4.1 Preliminary: comparison with levoglucosan concentrations

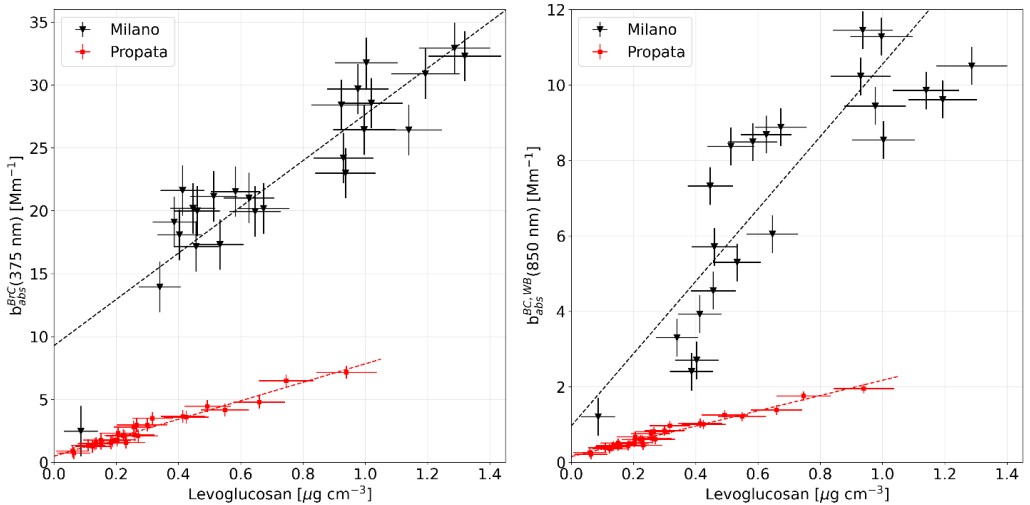

*Figure 2: Correlation between the levoglucosan concentration and the apportioned absorption coefficient of BrC at 375 nm (left*
*plot), and the absorption coefficient of BC due to wood burning at 850 nm (right plot). The rural site, Propata (red dots) exhibits a higher correlation than the urban site Milan (black triangles).*

To determine the goodness of the apportionment procedure even without the extra pre-processing step, default values for the free parameters were chosen ($\alpha_{BC} = \alpha_{FF} = 1$, $\alpha_{WB} = 2$) and the standard MWAA model for optical apportionment

(steps II and III above) was run. Figure 2 shows the correlation plots between the levoglucosan concentration and the relevant apportionment results, namely $b_{abs}^{BrC}(\lambda = 375\ nm)$ and $b_{abs}^{BC,WB}(\lambda = 850\ nm)$. The linear regression equations and coefficients of determination are given in Table 1.

**Table 1: Results of the correlation analysis between the apportioned data and the independent measurement, in this case**
**levoglucosan concentration. The goodness-of-fit of the correlation, represented by the coefficient of determination $R^2$, is much higher in Propata than in Milan.**

| Location | Absorption | Fit equation y = mx + q $\{m\} = Mm^{-1}\mu g^{-1}cm^3$   $\{q\} = Mm^{-1}$ | $R^2$ |
|---|---|---|---|
| Milan | $b_{abs}^{BrC}(\lambda = 375\ nm)$ | $m = 18.4\ \pm 1.8,$ $q = 9.3\ \pm 1.4$ | 0.82 |
| Milan | $b_{abs}^{BC,WB}(\lambda = 850\ nm)$ | $m = 9.6\ \pm 1.1,$ | 0.75 |





| | | | |
|---|---|---|---|
| | | $q = 1.0 \pm 0.9$ | |
| Propata | $b_{abs}^{BrC}(\lambda = 375\ nm)$ | $m = 7.3 \pm 0.3,$ $q = 0.48 \pm 0.10$ | 0.96 |
| Propata | $b_{abs}^{BC,WB}(\lambda = 850\ nm)$ | $m = 2.02 \pm 0.07,$ $q = 0.15 \pm 0.03$ | 0.97 |

**4.2 Analysis step I**

As described in Sect. 2, the first step of the upgraded apportionment model is to find the values of the absorption exponents that maximise the correlation between some of the model's output values and one or more independent techniques. In this

case, since the concentration of levoglucosan (hereinafter $c_l$) was measured on all samples, the following set of optimizations ('Levoglucosan' analysis preset in MWAA_MT) was carried out:

- vary $\alpha_{BC}$ to maximise the correlation between $b_{abs}^{BrC}(\lambda = 375\ nm)$ and $c_l$;
- vary $\alpha_{FF}$ to maximise the correlation between $b_{abs}^{BC,WB}(\lambda = 850\ nm)$ and $c_l$;

- vary $\alpha_{WB}$ to maximise the correlation between $b_{abs}^{BC,FF}(\lambda = 850\ nm)$ and $c_l$.

The resulting sets of exponents were $(\alpha_{BC},\ \alpha_{FF},\ \alpha_{WB})^P = (1.00 \pm 0.05;\ 1.00 \pm 0.02;\ 2.00 \pm 0.05)$ for Propata and $(\alpha_{BC},\ \alpha_{FF},\ \alpha_{WB})^M = (0.90 \pm 0.05;\ 0.90 \pm 0.02;\ 1.70 \pm 0.05)$ for Milan. Figure 3 shows the variation of the $R^2$ coefficient, in the two sites: the change of $\alpha_{WB}$ value does not produce any sizeable impact in the analysis of the rural

dataset whereas in Milan the best choice turned out to be $\alpha_{WB} = (1.70 \pm 0.05)$.

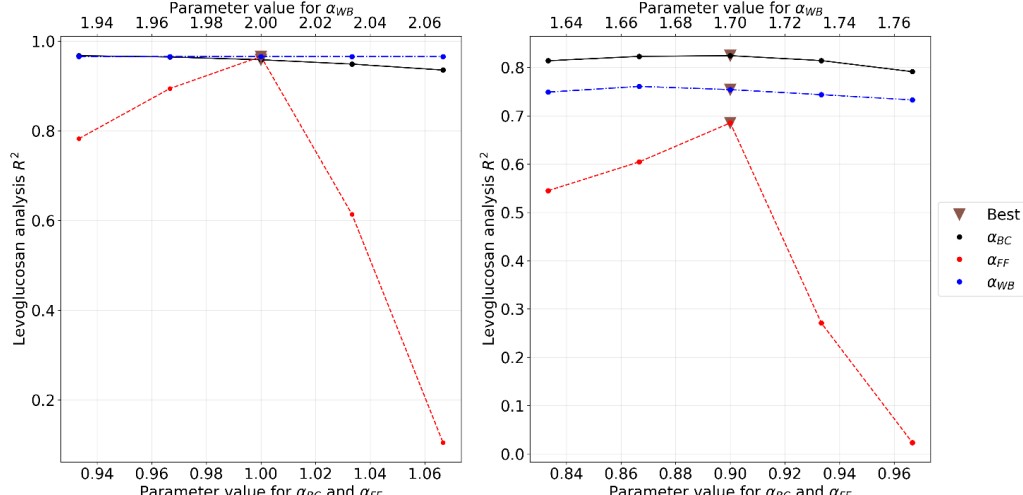

***Figure 3: Trend of the $R^2$ correlation coefficient between the levoglucosan concentration and the apportioned optical absorption coefficient vs. the value of the α exponents: Propata (left panel) and Milan dataset (right panel). The values of α parameter which***
***maximize the $R^2$ coefficient, are marked with a red X.***

This justifies the choice of setting $\alpha_{WB} = 2.0$ for the rural site, while the same choice is less robust for the urban site,



where $\alpha_{WB} = 1.7$ would be the more appropriate setting. The analysis confirms the usual choice of $\alpha_{BC} = 1.0$ for the rural site, while $\alpha_{BC} = 0.9$ gives the optimal value for the urban dataset, possibly indicating a further reprocessing and ageing of pure BC particles in the urban environment (Minderytė et al, 2022). As with $\alpha_{FF}$, the analysis yields $\alpha_{FF} = 1.0$ and $\alpha_{FF} = 0.9$ for the rural and urban site/dataset, respectively.

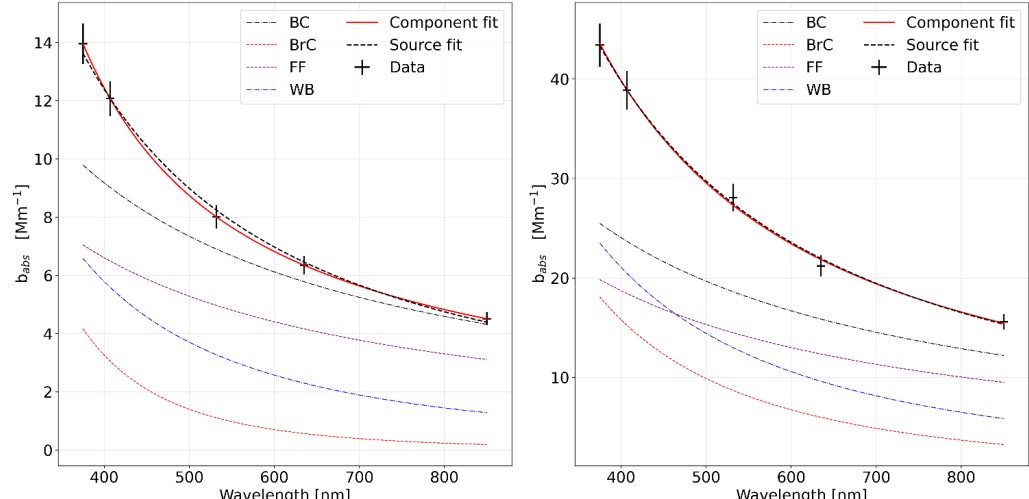

*Figure 4: Absorption coefficient plots for a representative sample from the rural site in Propata (left) and the urban site in Milan (right). The experimental data points are marked with uncertainty crosses. Superimposed on the graphs are the results of the fits carried out with the two different decompositions from Sect. 1.*


### 4.3 Analysis step II

The parameters found in Step I are then fixed for each dataset, and a complete double fit of the experimental data is performed for each sample following Eq. (1) and (2). An example of such fits for a sample belonging to each of the two datasets is shown in Fig. 4.





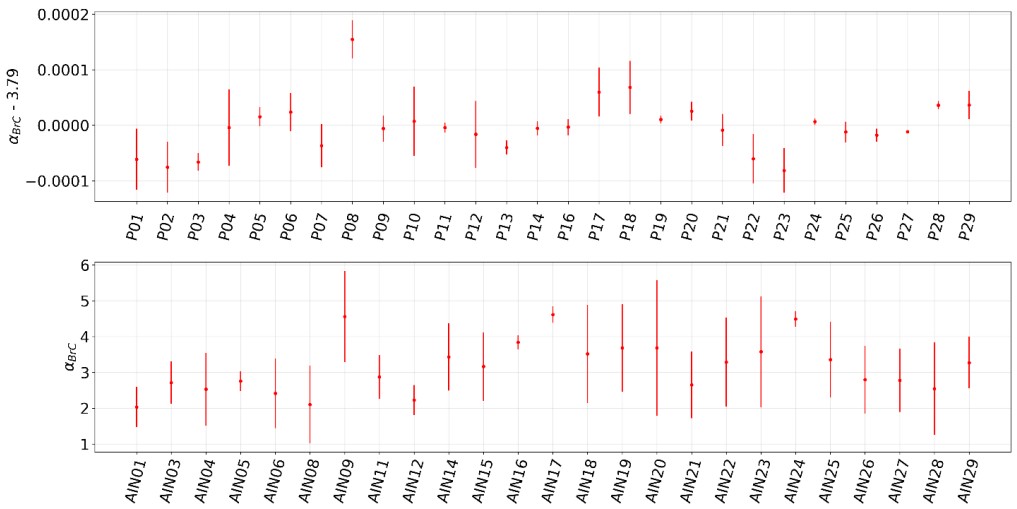


*Figure 5: Values of the parameter $\alpha_{BrC}$ for all the samples in the datasets: Propata/rural (top) and Milan/urban (bottom).*

One of the parameters determined during the fitting procedure is $\alpha_{BrC}$. Its range of variability over the entire dataset and the uncertainty associated with its value allow an estimation of the physical and chemical variability of the analysed

particulate. Figure 5 shows the determined values of $\alpha_{BrC}$ for the entire sampling period at the two sites. At a rural site such as Propata, where wood burning is the predominant source of carbonaceous particulate in the atmosphere during winter, BrC corresponds to a well-defined sub-category of organic carbon and its absorption properties are therefore constant. This is confirmed by the very small range of variation in the value of $\alpha_{BrC}$ obtained for all Propata samples, with $\alpha_{BrC}^{P} = 3.79 \pm 0.04$, where the uncertainty is due to inherent systematic variations in the minimization routine. At

the Milan urban site, the range of variability of $\alpha_{BrC}$ is much higher. This can be due to the fact that the urban aerosol is much more complex and contains a larger number of organic carbon compounds from different types of wood burned under different conditions, some of which could affect the optical behaviour of the aerosol. Moreover, stagnation conditions that favour ageing processes are typical of Milan (and the Po Valley in general): they generally lead to an increase in the molecular weight of organic matter, possibly enhancing light-absorption properties. Therefore, our model,

based on this simple decomposition, does not retrieve a sharp value for $\alpha_{BrC}$, since the optical properties of BrC vary strongly in Milan, unlike in Propata where the sampled particulate was comparatively simpler. The determined absorption exponent for BrC in Milan is therefore $\alpha_{BrC}^{M} = 3.5 \pm 1.1$.

### 4.4 Analysis step III

The optical apportionment of the absorption coefficient is performed for all available wavelengths according to Eq. (3).

Figures 6 and 7 show the apportioned $b_{abs}$ at UV (375 nm) and IR (850 nm) wavelengths for Propata and Milan, respectively. A general feature common to both datasets is the negligible absorption attributed to BrC at long wavelengths; this is consistent with previous work (Massabò et al., 2015).





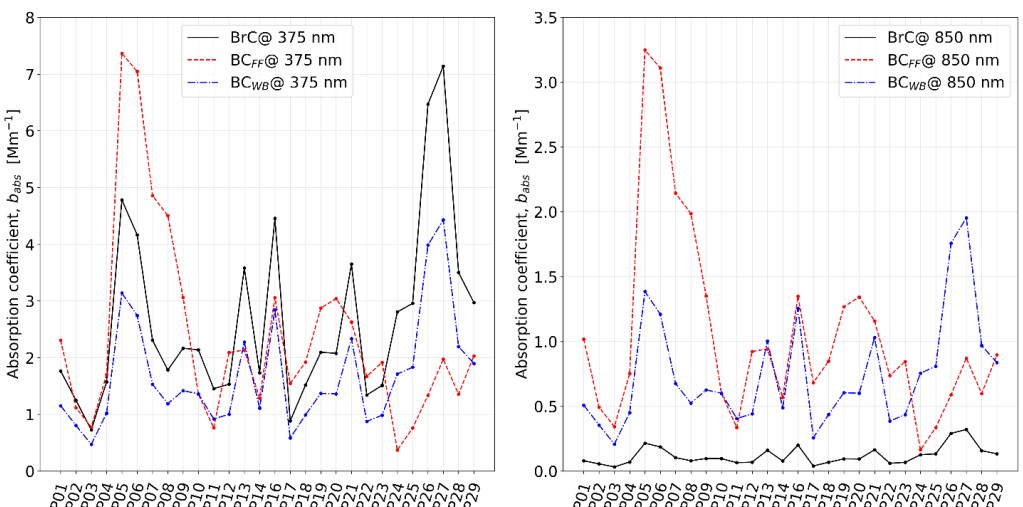

*Figure 6: Temporal variability of the apportioned absorption coefficients in Propata. Each sample covers a 48h period, starting from 07/11/2014 for P01.*

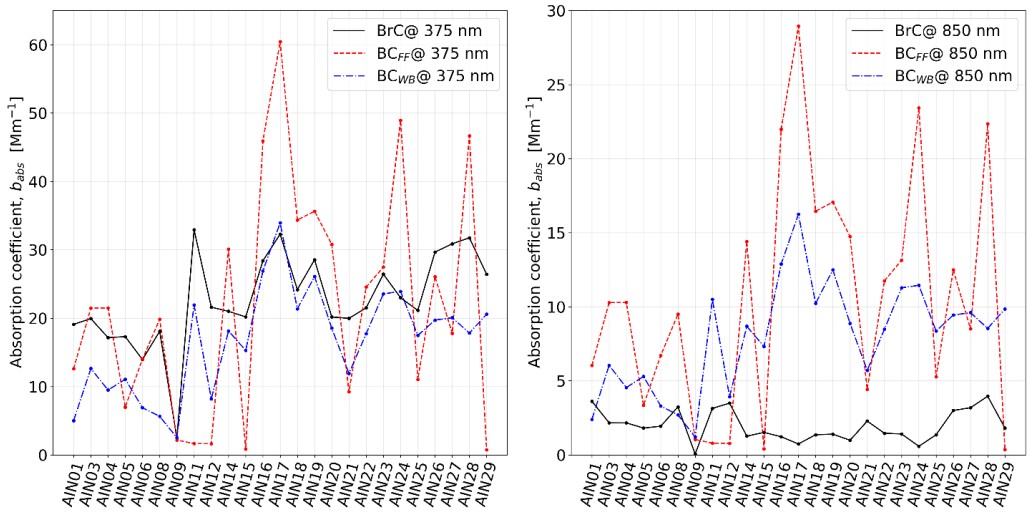

*Figure 7: Temporal variability of the apportioned absorption coefficients in Milan. Each sample covers a 12h period, starting from 21:00, 21/11/2016 for AIN01.*

### 4.5 Analysis step IV

Mass apportionment was performed for both datasets using the approach detailed in Sect. 2. The toolkit allows the user to choose to automatically determine the coefficients $k_1$ and $k_2$ from the linear regressions (see Eq. 4), or to set them manually. The latter approach is indicated when the dataset does not contain suitable candidate samples for the $k_1$



regression analysis (i.e. when there are no samples whose EC content is predominant as evidenced by an α close to 1), and especially when the values for $k_1$ and $k_2$ can be estimated by complementary methods or by previous analyses of aerosol samples taken at the same location during a comparable period of the year.

For this example application, the second approach was followed. For Milan, the regression coefficients were set to $k_1^M = 0.33\ g\ m^{-2}$, $k_2^M = 0.34\ g\ m^{-2}$, while for Propata they were set to $k_1^P = 0.24\ g\ m^{-2}$, $k_2^P = 0.35\ g\ m^{-2}$ as described in Bernardoni et al., 2017.

The average $EC_{FF}/EC$ ratio turned out to be  (49% ± 20%) and (58% ± 15%), and complementary $EC_{WB}/EC$ resulted (51% ± 20%) and (42% ± 15%), respectively in Milan and Propata. For the organic aerosol, the average $OC_{FF}/OC$ was found to be (25% ± 14%) and (18% ± 9%), while $OC_{WB}/OC$ was (58% ± 17%) and (61% ± 14%), in Milan

and Propata respectively. The non-combustion component of the organic aerosol, $OC_{NC}$, contributed (17% ± 15%) and (21% ± 15%) of the total $OC$  measured in Milan and Propata. For all the reported results, the uncertainty is understood as the standard deviation in the distribution of the mass-apportioned values of EC and OC for all the samples. These results are in full agreement with those reported in Bernardoni et al., 2017.

**5 Conclusions**

In the aerosol community concerned with aerosol source apportionment, the possibility of apportioning carbonaceous sources by exploiting optical properties has occupied much space in recent years. The main reasons for this growing interest are the diffusion of optical instruments that allow relatively easy-to-use, high-time resolution measurements. The main weakness of this apportionment methodology, based on optical measurements, is the practically obligatory choice of the critical parameters necessary as input, in particular $\alpha_{WB}$ and $\alpha_{FF}$, whose values vary considerably in the literature

(Sandradewi et al., 2008; Favez et al., 2010; Herich et al., 2011: Harrison et al., 2013; Massabò et al., 2015; Zotter et al., 2017; Forello et al., 2019). In this work, we show that it is possible to perform optical source and component apportionment of carbonaceous aerosols without constraining any physical parameters with *a priori* knowledge. Instead, the upgraded model presented here (MWAA_MT) allows the determination of these parameters for any specific receptor site, provided that a measurement using an independent technique able to trace biomass burning emissions is available

for comparison, also with different (lower) time resolution. This offers the advantage of an apportionment routine based entirely on experimental data, where computational parameters are automatically adjusted to best match the results with the data themselves. With this upgrade, it is possible to obtain α absorption exponents that are related to the specific site and season, allowing better characterization of future measurements at the same site or at sites with similarities (e.g., rural sites with similar geographical characteristics such as type of wood burnt). In addition, the α parameters obtained from

the analysis of robust, low-time resolution samples can be used to inform and fine-tune the apportionment procedure on high-time resolution data.

This "pre-processing step" has shown that, for the example dataset we considered, the values for $\alpha_{BC}$, $\alpha_{FF}$ and $\alpha_{WB}$ at a rural site are consistent with the literature, while in the case of an urban site, values of $\alpha_{BC} = 0.9$,  $\alpha_{FF} = 0.9$ and $\alpha_{WB} = 1.7$ seem to be a more appropriate choice.

We have also showed how sensitive the model is to the choice of some of these parameters: in our example, in particular, the choice of $\alpha_{FF}$ has the greatest impact on the reliability of the subsequent apportionment. It should be emphasised that this could be a feature of these specific data/sites: other datasets could be more sensitive to the variation of another one of the free parameters.  It is therefore recommended that, whenever possible, such an analysis be performed to determine the best value for any of these exponents.



Finally, we have described the operation of the new software toolkit, MWAA_MT, that we have used to perform this
       analysis and is made available to the scientific community.

**Author contribution**

TI and DM designed the extension to the original mathematical model. TI developed and tested the Python model code.
AB translated the software to R. TI prepared the manuscript with contributions from all co-authors.

**Competing interests**

The authors declare that they have no conflict of interest.

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
