# Peer review of "A new software toolkit for optical apportionment of carbonaceous aerosol"

_EGUsphere, 2023_

## Referee Comment (RC2)

General Comments:

The manuscript by Tommaso Isolabella et al. presented an upgrade to the Multi-Wavelength Absorbance Analyzer optical apportionment model. In addition to the apportionment of the absorption coefficient $b_{abs}$ in its components and sources, the extended model allows the retrieval of the Absorption Ångström Exponent of each component and source, thereby avoiding initial assumptions regarding these parameters. The deployment and application of this improved model toolkit holds some technical value.

Overall, the topic fits well within the scope of AMT. Before its publication, the following comments need to be addressed.

Specific Comments:
The parameter value for αWB varied from 1.94 (1.64) to 2.06 (1.76) in Fig. 3. Please explain reasons for choosing these ranges here. In addition, is there any specific reason that you used 0.02 as the interval in Fig. 3? The uncertainties caused by choosing different interval values and the ranges of parameter value for $\alpha_{WB}$ should be evaluated. Please elaborate.

The authors assumed that the absorption coefficient is decomposed into contributions from fossil fuel and wood burning, and that BrC is only produced by wood burning (Line 93-95). The authors need to address such uncertainties in the revised manuscript. In addition, such uncertainties should be evaluated at the different campaigns due to different primary emissions.

The authors compared the Propata campaign and Milan campaign datasets to verify whether the particulate sampled in a rural area has a different optical behavior than the aerosol sampled in an urban area. However, the comparisons have not been deeply discussed throughout this manuscript. For example, the differences between BrC and $BC_{WB}$ are similar across all sampling time in Propata, but the differences vary at different periods in Milan. Please elaborate.

In Section 4, a brief description of the Propata campaign and Milan campaign (including PM mass concentrations, composition, and sources) would be good. Otherwise, we don't know the general characterization of the two campaigns.

---

## Author Comment (AC2)

**Answers to RC1**

General Comments:

Optical apportionment of carbonaceous aerosol is an important process in the measurement of aerosol absorption properties. This manuscript presents an improved model without initial assumptions of parameters for distinguishing the composition and sources of light-absorbing carbonaceous aerosols based on the traditional optical apportionment model used for multi-wavelength absorption coefficient detection. From a scientific perspective, this study lacks significant innovation. Additionally, the deployment and application of this improved model toolkit holds some technical value. Therefore, it is recommended to reconsider the acceptance of this study after the following issues have been well solved.

*We thank the referee for her/his precious comments and suggestions. We would like to emphasize that indeed, the scope of our article is the description of a new software toolkit that implements an upgraded optical apportionment algorithm, of which we show the application and the potentiality to two case studies.*

Specific Comments:

1) The algorithm presented in this paper requires at least one additional independent measurement result (e.g., Levoglucosan), which significantly limits the application of this method. In Equations 1 and 2 within the text, each of them has four unknowns. In theory, the detection results from five wavelengths are sufficient to solve these unknowns. Why didn't the authors use the results from a multi-wavelength absorption analyzer for independent calculations?

*Even if using data at 5 wavelengths to fit 4 parameters is feasible in principle, problems arise from the point of view of numerical calculation since 2 of the four parameters to be fitted are exponents and the problem cannot be reduced to a linear model. Furthermore, from a mathematical point of view, **constraining some of these parameters is necessary**, since the functional form of Eqs. (1) and (2) is the same and fitting both equations would yield the same result if no parameters are fixed.*

*It is noteworthy that the software toolkit presented in this paper implements the original MWAA model, whereby an apportionment of optical absorption is achieved by fixing the Absorption Ångström Exponents for BC, FF and WB to predetermined values. This analysis can be carried out **without the need of any extra measurement**.*

*To better emphasize this concept, we propose to modify the text as following (lines from 70 in the introduction): "In this work we propose: 1) a toolkit that implements a revised version of the original MWAA model, as published in Massabò et al., 2015. This toolkit has been rewritten and optimized in Python and R, and is available for use by the scientific community. It has been also extended with the possibility to use an arbitrary number of spectrally resolved absorption coefficients, as long as at least 5 wavelengths are available. This model is self-consistent and can be applied to purely optical data without the need of any other information. 2) An upgrade to the original MWAA model that directly allows source and component apportionments of absorption data without the need to set any parameters before running the model. This is achieved by performing the apportionment analysis along with a correlation study with independent measurements such as chemical speciation or elemental composition. The parameters are then automatically set by the algorithm, based on the values that give the best correlation with the independent measurements. The new software toolkit presented here is written in two of the most widely used scientific programming languages, Python and R, to perform this analysis automatically. Output of the presented toolkit (MWAA_MT) are the following quantities: $\alpha_{FF}$, $\alpha_{WB}$, $\alpha_{BC}$, $\alpha_{BrC}$, and the carbonaceous masses for fossil fuels and wood burning: $EC_{FF}/OC_{FF}$, and $EC_{WB}/OC_{WB}$, respectively, where EC (OC) stands for Elemental (Organic) Carbon. Finally,*

*to demonstrate the capability of the upgraded model, we provide an example application to data published elsewhere (Bernardoni et al., 2017)."*

However, when an independent measurement is available (as an example, Levoglucosan concentration or the $^{14}C/^{12}C$ ratio), the software toolkit allows an upgraded version of the MWAA model whereby the fit parameters are adjusted with the aim of maximising the correlation of the optical apportionment with the independent measurement. With this in mind, the additional data required to use the upgraded model allows for a more complete optical characterization of the aerosol. Furthermore, the additional measurement can be performed on a subset of the entire data, just to find the right parameters for the model that can be subsequently applied to a larger dataset and/or with different time resolution.

2) In the algorithm described in this paper, αBC, αFF, and αWB remain constant over a certain period of time (such as during a field experiment), while αBrC varies with time. This is not reasonable. For example, in the observation example in Milan, αBrC clearly varies with time, while αWB is assumed to be a constant value in this period. This can introduce significant errors into the calculations. For example, in Figure 6, the trends of BrC and BCWB are nearly identical, while in Figure 7, there is a significant difference between them. This distinction may be a result of the algorithm rather than the environmental conditions themselves.

$\alpha_{BC}$ and $\alpha_{BrC}$ are intrinsic properties of the respective aerosol species, while $\alpha_{FF}$ and $\alpha_{WB}$ are intrinsic properties of the respective aerosol sources. Since BC is a relatively well-defined and well-characterized species, it is reasonable to assume that its AAE will remain approximately constant over a measurement campaign. However, BrC is a much more variable and complex category of aerosol. Its AAE is expected to have a higher variance, even within the same measurement campaign, especially in a complex urban site like Milan where a complex mixture of different sources is expected, and strong aging processes occur. Calculating $\alpha_{BrC}$ as proposed by the MWAA model allows to catch this variability and is, therefore, the key aspect of the model.

As for $\alpha_{FF}$ and $\alpha_{WB}$, they are assumed to be constant by the original Aethalometer model (Sandradewi et al., 2008) and we have simply integrated it to our component apportionment model to reach a source-component apportionment. We agree with the reviewer that the 2-source model with constant optical properties can have higher limitations, especially in cases where there are more sources, or modification of aerosol by WB due to aging. The assumption is more accurate in a simple rural site, such as Propata, than in a busy and polluted urban site such as Milan. However, the AAE values retrieved through this method can be seen as campaign-averages that are representative of the specific sources present in that location in that season. The assumption of having a fixed $\alpha_{WB}$ is especially robust in cases where the aerosol in the area is mainly primary and no further BrC sources/processes are expected. This means that most of the BrC is produced via WB, as is the case of Propata. In fact, in Propata, $\alpha_{BrC}$ does not vary, and correspondingly, BrC and BC$_{WB}$ correlate very well. The same can't be said about Milan, where the particulate is impacted by a number of different sources, and it is heavily processed due to stagnation.

We propose to insert the following text in line 251: "*The main difference between the two sites is the correlation between $b_{abs}^{BrC}$ and $b_{abs}^{BC,WB}$. In Propata (Fig. 6) the correlation is high, as can be inferred by the blue and black lines having the same time trend. This means that most of the BrC is produced via WB. On the other hand, in Milan (Fig. 7) this correlation is lower, and BrC cannot be entirely attributed to WB. In fact, in Milan the particulate is impacted by a number of different sources, and it is heavily processed due to stagnation.*"

3) In Figure 3, in the left panel, αBC has a higher R2 value around 0.93, while in the right panel, αWB has a higher R2 value around 1.67. Why not use these two values as fitting results?

We introduced a tolerance parameter Δ to improve the stability of the model and eliminate physically insignificant $R^2$ fluctuations. The idea is that if the difference between the $R^2$ for a given α and the $R^2$ for the current best α is smaller than Δ, then the variation is not considered significant and the current best α value is retained. For the analysis presented in the paper, Δ was set to 0.01 (v. lines 120-122 and Fig. 4).

In case we run the analysis with the $\alpha_{BC}$=0.93 in Propata, the apportionment results are quite similar (differences are below 8%). However, we choose $\alpha_{BC} = 1$ since the $R^2$ is almost identical, and also because the $\alpha_{BC}$ sweep analysis in Propata confirms it as the most sensible choice (please refer to the supplementary material). Similarly, if we run the analysis with $\alpha_{WB}$= 1.67 in Milan, the apportionment results vary by up to 7%, and the correlation between $BC_{WB}$ and BrC remains the same. On account of these consideration, we considered the changes in $R^2$ between different α values to be significant only if higher than Δ = 0.01.
We want to emphasize that the choice of Δ may vary depending on the characteristics of the samples under scrutiny. The user may change the resolution value (Δ) according to her/his scientific judgement, down to a value of 0 meaning that any α value maximizing the relevant $R^2$ will be chosen, regardless of the physical meaning of its significant digits.

4) In P6L178, the author mentioned that the Milan campaign had 20 samples, but in Figure 5, there are 25 samples. Why is that?

Thank you for your correction. We changed the text and now P6L178 reads "[…] and 25 samples from the Milan campaign ("AIN" samples)."

5) The abbreviations BC, BrC, FF, and WB should only be introduced with their full names the first time they appear, and there's no need to reintroduce them later (e.g., as in P3L90). Similarly, EC should be introduced with its full name the first time it appears.

Thank you for pointing it out. We corrected the text accordingly, in P3L90 and P3L94. Additionally, P2L77-78 now read "carbonaceous masses for fossil fuels and wood burning: $EC_{FF}/OC_{FF}$, and $EC_{WB}/OC_{WB}$, respectively, where EC (OC) stands for Elemental (Organic) Carbon."

REFERENCES

[1] Massabò et al., *Multi-wavelength optical determination of black and brown carbon in atmospheric aerosols,* 2015, Atmos. Env. 108, 1-12

---

## Author Response (AR1)

**Answers to RC1**

General Comments:

Optical apportionment of carbonaceous aeroso" is an important process in the measurement of aerosol absorption properties. This manuscript presents an improved model without initial assumptions of parameters for distinguishing the composition and sources of light-absorbing carbonaceous aerosols based on the traditional optical apportionment model used for multi-wavelength absorption coefficient detection. From a scientific perspective, this study lacks significant innovation. Additionally, the deployment and application of this improved model toolkit holds some technical value. Therefore, it is recommended to reconsider the acceptance of this study after the following issues have been well solved.

We thank the referee for her/his precious comments and suggestions. We would like to emphasize that indeed, the scope of our article is the technical presentation of a software toolkit that implements an upgraded optical apportionment algorithm, of which we show the application to two case studies.

Specific Comments:

1) The algorithm presented in this paper requires at least one additional independent measurement result (e.g., Levoglucosan), which significantly limits the application of this method. In Equations 1 and 2 within the text, each of them has four unknowns. In theory, the detection results from five wavelengths are sufficient to solve these unknowns. Why didn't the authors use the results from a multi-wavelength absorption analyzer for independent calculations?

The additional independent measurement, such as Levoglucosan concentration or $^{14}C/^{12}C$ ratio, allows an extension of the original MWAA model, presented in [1]. The base model, represented by Equations (1) and (2) within the text, assumes fixed values for three of the four Absorption Ångström Exponents that appear in the equations. From a mathematical perspective, constraining the range of these parameters is necessary, since the functional form of Eqs. (1) and (2) is the same, and fitting both would yield the same result when the parameters are not fixed. The software toolkit presented in this paper implements the original MWAA model, whereby an apportionment of optical absorption is achieved by fixing the Absorption Ångström Exponents for BC, FF and WB to predetermined values. This analysis can be carried out **without the need of any extra measurement**. However, if an independent measurement is available, the software toolkit allows to employ an upgraded version of the MWAA model whereby the fit parameters are adjusted with the aim of maximising the correlation of the optical apportionment with the independent measurement. In this light, the additional data necessary to utilize the upgraded model give access to a more complete optical characterization of the aerosol. Furthermore, the additional measurement can be done on a subset of the entire data, just to find the right parameters for the model that can be subsequently applied to a larger dataset and/or with different time resolution.

2) In the algorithm described in this paper, αBC, αFF, and αWB remain constant over a certain period of time (such as during a field experiment), while αBrC varies with time. This is not reasonable. For example, in the observation example in Milan, αBrC clearly varies with time, while αWB is assumed to be a constant value in this period. This can introduce significant errors into the calculations. For example, in Figure 6, the trends of BrC and BCWB are nearly identical, while in Figure 7, there is a significant difference between them. This distinction may be a result of the algorithm rather than the environmental conditions themselves.

$\alpha_{BC}$ and $\alpha_{BrC}$ are intrinsic properties of the respective aerosol species; $\alpha_{FF}$ and $\alpha_{WB}$ are intrinsic properties of the respective aerosol sources. BC is a relatively well-defined and well-characterized species, with simple

optical properties and therefore it is reasonable to assume its AAE will remain approximately constant over a measurement campaign. The same can't be said about BrC which is a much more variable and complex category of aerosol. Its AAE will by definition have a higher variance, even within the same measurement campaign, especially in a complex urban site like Milan. As for $\alpha_{FF}$ and $\alpha_{WB}$, it is reasonable to assume that in a specific location and season, the aerosol sources will remain approximately the same and therefore the associated AAE will not change much. This is, of course, more accurate in a simple rural site such as Propata than a busy, polluted urban site such as Milan. Still, the AAE values retrieved through this method can be seen as campaign-averages that are representative of the specific sources present in that location in that season. The assumption of having a fixed $\alpha_{WB}$ and a variable $\alpha_{BrC}$ is especially robust in cases where the aerosol in the area is mainly primary. This means that most of the BrC is produced via WB, as is the case of Propata. In fact, in Propata, $\alpha_{BrC}$ does not vary and correspondingly, BrC and BC_WB correlate very well. The same can't be said about Milan, where the particulate is heavily processed due to stagnation.

3) In Figure 3, in the left panel, αBC has a higher R2 value around 0.93, while in the right panel, αWB has a higher R2 value around 1.67. Why not use these two values as fitting results?

We introduced a tolerance parameter Δ to improve the stability of the model and rule out insignificant $R^2$ fluctuations. In particular, if the difference between the $R^2$ for a given α and the $R^2$ for the current best α is smaller than Δ, then the variation is not considered significant and the current best α value is retained. For the analysis shown in the paper, Δ was set to 0.01 (v. lines 120-122 and Fig. 4). If we run the analysis with the $\alpha_{BC}=0.93$ in Propata, the apportionment results are quite similar (differences up to 8%), but we are drawn to choose $\alpha_{BC} = 1$ since the $R^2$ is practically identical, and also because of the $\alpha_{BC}$ sweep analysis that, in Propata, confirms $\alpha_{BC} = 1$ is the most sensible choice (please see supplementary material).  Similarly, if we run the analysis with $\alpha_{WB}= 1.67$ in Milan, the apportionment results vary by up to 7%, and the correlation between BC_WB and BrC doesn't change. On account of these consideration, we evaluated the changes in $R^2$ between different α values to be significant only if higher than Δ = 0.01. We emphasize that this choice of Δ might depend upon the characteristics of the samples under scrutiny. The resolution value (Δ) can be changed by the user according to their scientific judgement, down to a value of 0 meaning any α value maximizing the relevant $R^2$ will be chosen, irrespective of the physical meaning of its significant digits.

4) In P6L178, the author mentioned that the Milan campaign had 20 samples, but in Figure 5, there are 25 samples. Why is that?

Thank you for your correction. We changed the text and now P6L178 reads "[…] and 25 samples from the Milan campaign ("AIN" samples)."

5) The abbreviations BC, BrC, FF, and WB should only be introduced with their full names the first time they appear, and there's no need to reintroduce them later (e.g., as in P3L90). Similarly, EC should be introduced with its full name the first time it appears.

Thank you for pointing it out. We corrected the text accordingly, in P3L90 and P3L94. Additionally, P2L77-78 now read "carbonaceous masses for fossil fuels and wood burning: EC_FF/OC_FF, and EC_WB/OC_WB, respectively, where EC (OC) stands for Elemental (Organic) Carbon."

REFERENCES

[1] Massabò et al., *Multi-wavelength optical determination of black and brown carbon in atmospheric aerosols,* 2015, Atmos. Env. 108, 1-12

**Answers to RC2**

General Comments:
The manuscript by Tommaso Isolabella et al. presented an upgrade to the Multi-Wavelength Absorbance Analyzer optical apportionment model. In addition to the apportionment of the absorption coefficient b abs in its components and sources, the extended model allows the retrieval of the Absorption Ångström Exponent of each component and source, thereby avoiding initial assumptions regarding these parameters. The deployment and application of this improved model toolkit holds some technical value. Overall, the topic fits well within the scope of AMT. Before its publication, the following comments need to be addressed.

We would like to thank the referee for the time she/he invested in reviewing our article, and for the stimulating comments and precious suggestions.

Specific Comments:
The parameter value for $\alpha_{WB}$ varied from 1.94 (1.64) to 2.06 (1.76) in Fig. 3. Please explain the reasons for choosing these ranges here. In addition, is there any specific reason that you used 0.02 as the interval in Fig. 3? The uncertainties caused by choosing different interval values and the ranges of parameter value for $\alpha_{WB}$ should be evaluated. Please elaborate.

Thank you for pointing this out. We propose to add a clarification in the caption to Fig. 3, reporting: "*The plots are shown only for the last iteration of the preprocessing step.*"

The plots refer only to the last preprocessing iteration. To elaborate on the matter, the ranges for each $\alpha$ parameter are updated dynamically at each iteration of the preprocessing stage. The default starting ranges for $\alpha_{FF}$ and $\alpha_{BC}$ are set as [0.8, 1.2], and for $\alpha_{WB}$, it is [1.8, 2.2], each with a step size of 0.1. After each iteration, the new best triplet for $\alpha_{WB}$, $\alpha_{FF}$ and $\alpha_{BC}$ is found, a new (narrower) range is computed around each of the best values for the new alphas, and a new (smaller) step size is chosen. For instance, let's assume the first iteration results in the triplet (0.9, 0.8, 2.0). Then the new search intervals will be [0.80, 1.00], [0.70, 0.90], [1.90, 2.10] and the new step size will be 0.05. The process continues for all the subsequent iterations.
Following the advice of the referee, we propose to add the following lines starting from line 210: "*Through the sensitivity tests we performed on the preprocessing step, we discovered that the apportioned optical absorption coefficients can vary by up to 10% by adjusting the values of the $\alpha$ parameters within their uncertainty brackets. We estimated the uncertainty of the $\alpha$ parameters by considering the steepness of the $R^2$ vs. $\alpha$ curves. The curve of $\alpha_{FF}$ is very steep, which led us to estimate an uncertainty of 0.02, whereas the $R^2$ vs $\alpha$ curves for the other two parameters were flatter, indicating a larger uncertainty for these parameters.*"

The authors assumed that the absorption coefficient is decomposed into contributions from fossil fuel and wood burning, and that BrC is only produced by wood burning (Line 93-95). The authors need to address such uncertainties in the revised manuscript. In addition, such uncertainties should be evaluated at the different campaigns due to different primary emissions.

We appreciate this comment; we propose to remove the simplifying assumption made in line 92, which only considered WB as the source of BrC. In general, BrC can have other sources (see for example [1]). Furthermore, atmospheric mixing and processing can alter its optical properties over time. Therefore, the amount of BrC in the sample may be underestimated. Our model searches for an aerosol component *with the optical properties* of BrC (i.e. a high AAE); how much of it is actually BrC depends on a number of factors. The correlation between BrC and levoglucosan concentration is a good indicator of wood being the main source of BrC. If the correlation is low, the model may have found BrC from other sources than WB, or the BrC produced via WB may have degraded through atmospheric aging and now exhibits different optical properties. For example, in Milan, where there are multiple sources of carbonaceous aerosol, the model struggles to distinguish between the sources. Thus, the correlation between the apportioned coefficients in Milan and the concentration of levoglucosan (Fig. 2, black triangles and Table 1) is lower than in Propata ($R^2$=0.82), indicating that the model retrieves only a part of the emissions due to wood burning in the urban site. We revised the final part of the conclusion, and in particular we propose to add the following consideration at line 305: "*We would like to underline that the Milan case study is to be considered as a stress test of our algorithm: the context is very complex due to the presence of a large number of sources such as traffic, biomass combustion, industry, etc., in a city with over 1.3 million inhabitants. The city is also subject to major regional transport events, high PM concentrations (average PM10 value during the campaign of 68.3 $\pm$ 25.6 $\mu g\ m^{-3}$) and air stagnation conditions resulting in a high level of aerosol reprocessing. On the other hand, when it comes to the Propata dataset, the correlation with levoglucosan is much higher ($R^2$=0.96), indicating that within the experimental uncertainties the assumption that BrC is only produced by WB is satisfied.*"

Last, it is possible that the assumption of two-component apportionment is not suitable at the sampling site due to the poor correlation between BrC and levoglucosan. This could be due to the role of some types of mineral dust in light absorption. However, it is important to note that the impact of mineral dust can usually be considered negligible at our latitudes since it occurs only occasionally and for very limited periods.

The authors compared the Propata campaign and Milan campaign datasets to verify whether the particulate sampled in a rural area has a different optical behavior than the aerosol sampled in an urban area. However, the comparisons have not been deeply discussed throughout this manuscript. For example, the differences between BrC and BC WB are similar across all sampling time in Propata, but the differences vary at different periods in Milan. Please elaborate.

The goal of this work is to compare the optical behaviour of the aerosol in the urban and rural with the objective of carrying out an optical apportionment. The apportionment model aims to establish a correlation between the aerosol composition and its sources based on its particular optical behaviour. As already mentioned, as for the different correlation between BrC and $BC_{WB}$ in Propata and in Milan, the discrepancy is because the aerosol in Propata is mostly primary and therefore the BrC found in this site is mainly due to WB. For this reason, BrC and $BC_{WB}$ in Propata correlate very well between each other, and with the levoglucosan concentration. On the contrary, the aerosol in Milan is heavily reprocessed, resulting in a weaker correlation.

In Section 4, a brief description of the Propata campaign and Milan campaign (including PM mass concentrations, composition, and sources) would be good. Otherwise, we don't know the general characterization of the two campaigns.

A description of both measurement campaigns has already been presented in the paper Bernardoni et al., 2017. Actually, no information on PM composition is available except Levoglucosan concentrations; in

consideration of the referee's suggestion, we decided to add in the text the average value of the PM concentration at both sites. At line 176 we propose to add a sentence as follows: "*No information on chemical speciation (except Levoglucosan) was available at the two sites; the average PM10 concentration measured at Propata and Milan was 8.3 $\pm$ 6.0 $\mu$g m$^{-3}$ and 68.3 $\pm$ 25.6 $\mu$g m$^{-3}$, respectively*".

**References**

[1] Corbin, J. C. et al. (2018). *Brown and black carbon emitted by a marine engine operated on heavy fuel oil and distillate fuels: Optical properties, size distributions, and emission factors*, Journal of Geophysical Research: Atmospheres, 123, https://doi.org/10.1029/2017JD027818

[2] Bernardoni, V., Pileci, R. E., Caponi, L., Massabò D. (2017): The Multi-Wavelength Absorption Analyzer (MWAA) Model as a Tool for Source and Component Apportionment Based on Aerosol Absorption Properties: Application to Samples Collected in Different Environments, Atmosphere, 8(11), :218, https://doi.org/10.3390/atmos8110218, 2017.